# Comparing Logistic Regression Models with Alternative Machine Learning Methods to Predict the Risk of Drug Intoxication Mortality

**DOI:** 10.3390/ijerph17030897

**Published:** 2020-01-31

**Authors:** YoungJin Choi, YooKyung Boo

**Affiliations:** 1Department of Healthcare Administration, Eulji University, Seongnam 13135, Korea; 2Department of Health Administration, Dankook University, Cheonan 31116, Korea

**Keywords:** drug intoxication, influencing factor, logistic regression, machine learning, mortality prediction

## Abstract

(1) Medical research has shown an increasing interest in machine learning, permitting massive multivariate data analysis. Thus, we developed drug intoxication mortality prediction models, and compared machine learning models and traditional logistic regression. (2) Categorized as drug intoxication, 8,937 samples were extracted from the Korea Centers for Disease Control and Prevention (2008-2017). We trained, validated, and tested each model through data and compared their performance using three measures: Brier score, calibration slope, and calibration-in-the-large. (3) A chi-square test demonstrated that mortality risk statistically significantly differed according to severity, intent, toxic substance, age, and sex. The multilayer perceptron model (MLP) had the highest area under the curve (AUC), and lowest Brier score in training and validation phases, while the logistic regression model (LR) showed the highest AUC (0.827) and lowest Brier score (0.0307) in the testing phase. MLP also had the second-highest AUC (0.816) and second-lowest Brier score (0.003258) in the testing phase, demonstrating better performance than the decision-making tree model. (4) Given the complexity of choosing tuning parameters, LR proved competitive when using medical datasets, which require strict accuracy.

## 1. Introduction

The motive for consuming toxic substances may vary according to its social or regional background and is influenced by generational changes and cultural development. Particularly, in Korea, which has undergone rapid social, economic, and cultural developments, drug-induced suicide attempts and cases of drug intoxication from drug abuse are rising [1]. Intoxication can be defined as a state inducing a functional or structural impairment of the body as a result of exposure to a natural or synthetic substance [2]. Although the rate of intoxication mortality may be lower than that of other injuries, the scale of socioeconomic damage incurred by intoxication is surmised to be substantial, taking into consideration a patient’s rehabilitation process in addition to the cost of treatment [3,4].

Drug intoxication refers to an impairment or adverse reaction caused by an intentional or accidental drug overdose over a short period of time [5]. The clinical symptoms vary widely depending on the cause of poisoning and type of substance. Some cases may be fatal and untreatable [6]. In addition, it poses a significant medical and societal problem, as drug intoxication induces serious complications, and inflicts substantial economical and mental suffering on caregivers and patients alike, even after treatment [7,8]. To attenuate the aggravating problem of intoxication, information on the current state of intoxication is needed in addition to further promotion of preventive efforts, involving prompt and accurate treatment, to reduce anticipated sequela and complications after treatment.

Today, healthcare institutions worldwide have developed a variety of mortality prediction models, such as Acute Physiology and Chronic Health Evaluation, Simplified Acute Physiology Score, and Mortality Probability Model, to enhance the quality of medicine [9,10] and the validity and fit of these models. Particularly, artificial intelligence-based prediction models for inpatient mortality, 30-day unplanned readmission, long lengths of stay, and discharge diagnoses have also been developed [11,12]. Regarding acute trauma mortality prediction models, one study developed a model using a trauma and injury severity score based on the severe trauma database of regional emergency medical centers^,^ [13].

Most past studies have researched drug intoxication prediction models based on the severity and incidence. The development of risk prediction models is challenging as many factors, including the aforementioned considerations, should be examined [14,15,16]. Thus, the development usually involves a combination of numerous diagnostics, and performance statistics for model calibration and discrimination [17].

Hippisley-Cox et al. developed a model to predict mortality risk for acutely intoxicated inpatients who visited hospital as unplanned or emergency admissions [18]. The variables employed in this model were patients deceased during admission, the identified substances, and patient’s age, and gender. The controversial issue, related to the risk prediction model, is associated with the ways of finding and applying the best predicting models for the situation. There are two approaches for choosing a predicting model: The traditional and the machine learning models recently used in the field, including a decision tree (DT) and multilayer perceptron (MLP).

Logistic regression (LR) is a traditional model commonly employed in medical applications to interpret clinical data in depth. On other hand, the machine learning models recently used in the field, including a DT, support vector machine, and MLP, have been widely used as of late [19,20,21,22,23,24,25,26,27]. Herein, the LR and the intensive approaches (DT and MLP) were compared to identify the best performing prediction model in medical mortality [28,29]. For the comparison, an external validation was used to investigate the predictions made by these models in terms of calibration and discrimination.

## 2. Materials and Methods

### 2.1. Data Set

Since 2005, the Korea Centers for Disease Control and Prevention conduct a Korean National Hospital In-depth Injury Survey on discharged patients in 170 hospitals nationwide that contain 100 beds or more. In this study, from the 2,149,572 samples within the Korea National Hospital Discharge Survey (KNHDS) from 2008 to 2017 (10 years), 8,937 samples, categorized as drug intoxication (ICD-10 codes T36.0–T65.9), were used to predict the mortality risk among inpatients with injuries as shown in Figure 1.

To identify the factors that affect the mortality due to drug intoxication, the dependent variable was set to mortality, while the independent variables were primarily set to those mentioned by previous studies, like as sex, age, toxic substance, intent of intoxication, severity, area of residence, and risk factor method of payment, method of hospital admission, and mental disorder. An importance analysis was conducted on these independent variables to identify which should be used in the model comparisons. Based on the results, the following five variables with an importance of 10% or higher, were selected: age, toxic substance, intent of intoxication, severity, and risk factor.

Age, which is a continuous variable, was adjusted to an ordinal variable by dividing the age groups into 65 years or older and below 65 years. Severity was assessed based on the Charlson comorbidity index score, a widely used index for the adjustment of comorbidities calculated by the sum of weighted scores based on the presence/absence of 19 different medical conditions. In addition, toxic substance was grouped into toxic drugs, alcohol, environmentally harmful substances, and other substances. Intent of intoxication was divided into intentional and unintentional, and risk factors were divided into conflict with relatives, physical illness, mental illness, financial problem, and other factors.

### 2.2. Methods

The calibration is one of the most important factors in prediction, as it is closely related to the model’s reliability [22]. The calibration was performed both internally and externally. Internal validation involves split of the surveyed data into training, validation, and testing sets. Then, external validation, which is a key step before clinical usage, involves model calibration and discrimination investigating the performance of the prediction in different, but possibly related setting [23].

In this study, three performance measures were used to assess model performance: Brier score, area under the Receiver Operating Characteristic (ROC) curve (AUC), and calibration-in-the-large [22]. Overall performance compares the predicted outcome with the actual outcome. Nagelkerke’s R^2^ and the Brier score are frequently used. The Brier score utilizes the mean square error between predicted probabilities and the expected values and is modified to be applied to logistic regression. The equation for the Brier score is shown below. The score ranges from 0 to 0.25 and a score closer to 0 indicates better model performance.
Brier = mean((y−p)^2^) = mean(y x (1−p)^2^ + (1−y) x p^2^)(1)

Second, discrimination represents whether the model is capable of distinguishing people with and without an illness. Discrimination slope, Concordance index (C index), and AUC are often used. In the present study, AUC, area under a ROC curve, was used. Calibration represents the concordance between the predicted and actual values. The Hosmer-Lemeshow’s goodness of fit test and calibration-in-the-large are generally used, and we opted for the latter in this study. It is a measure of the difference between the mean observed value and the mean predicted value in linear regression that is modified to apply in LR using logit functions. It computes the difference in the log odds of the actually observed value and the predicted value. A value closer to 0 indicates better performance.
(2)$$\mathrm{Calibration}-\mathrm{in}-\mathrm{the}-\mathrm{large}(a)\;=\;\mathrm{logit}(y\;=\;1)\;-\;\mathrm{logit}(p)\;=\;\ln(\mathrm{odds}(\mathrm{mean}(y))\;-\;\ln(\mathrm{odds}(\mathrm{mean}(p))\;=\;\ln(\frac{mean\left(y\right)}{1-mean\left(y\right)})\;-\;\ln(\frac{mean\left(p\right)}{1-mean\left(p\right)})$$

## 3. Results

### 3.1. Statistical Analysis

This study analyzed the factors that affect mortality from drug intoxication with chi-square tests and compared the performances of prediction models using the IBM SPSS Statistics 25 (IBM Corp. Released 2017. IBM SPSS Statistics for Windows, Version 25.0. Armonk, NY: IBM Corp.). To test the performance and validity of the prediction models, we divided the samples into training, validation, and testing phases. The sample distribution per phase is shown in Table 1.

A total of 4,442 samples over five years, from 2008 to 2012, 2,575 (96.4%) survived drug intoxication. A chi-square test was performed on the drug intoxication mortality predictors, namely age, toxic substance, severity, cause of risk, mental and intent, and the results showed that mortality significantly differed in most of these variables. However, the discrepancy according to cause of risk was significant at 0.05 in all three phases (Table 1).

To avoid overfitting and evaluate exact prediction performance, the dataset was split into training, validation, and a separated test set [30]. The prediction performance and ROC plots of three datasets are demonstrated in Table 2.

To compare, we created box graphs as shown in Figure 2 below, for the training, validation, and testing phases in the three prediction models, with the mortality prediction as the *y*-axis and the actual treatment outcome (0: Death 1: Survival) as the *x*-axis. In terms of predicting death (0 on *x*-axis), all three models had many outliers (O) that deviated from the quartiles. However, the DT model had relatively fewer outliers compared to the LR and MLP models. On the other hand, in terms of predicting survival (1 on *x*-axis), fewer outliers appeared beyond the quartiles than those for death prediction. In the validation and testing phases of the DT model, the variance was large, but the values were within 50% of the center. In other words, the DT is the superior model in predicting mortality, while the LR model has a relatively smaller variance than others in predicting survival.

### 3.2. Modeling Results

In the study, seven years of drug intoxication data were divided into training and validation data, and the models’ performances were tested using three years of data. We predicted the risk of mortality using the following modeling approaches: LR, DT, and MLP.

The MLP trains the network by error back-propagation, and with one hidden layer with nine neurons. The hyperbolic tangent activation function was used, and softmax was used as the output layer function for the prediction of diseases [31]. The sensitivity analysis was performed with neurons. When increasing neurons up to nine, the prediction accuracy increased and did not improve considerably afterwards. As such, nine neurons were used as to avoid an over-fitting problem [32]. In order to train the MLP, the scaled conjugate gradient algorithm was used, which tends to minimize error between the output of an MLP and the desired output [33].

The goal of DT was to produce subsets of the data which were as homogeneous as possible with respect to the target variable. In this study, we used the measure of gini impurity that is used for categorical variables [34]. In addition, the minimum parent size was set at 100, minimum child size was 50, and minimum improvement was 0.0001.

For the performance testing, the Brier score was used to compare overall performance, and the AUC to analyze discrimination. Further, calibration-in-the-large was used to analyze calibration. The model performance statistics are shown in Table 2 and plotted in Figure 3.

In the training phase, the AUC ranged from 0.779 to 848. MLP had the highest AUC (0.848), while LR had the lowest (0.779). The Brier score ranged from 0.0599 to 0.0604. MLP had the lowest Brier score (0.0599), and DT had the highest (0.0604). The absolute values of calibration-in-the-large ranged from 0.0034 to 0.3186. LR had the lowest score (0.0034) and DT had the highest (0.3186). In the validation phase, all methods had an AUC range between 0.788 and -0.853, and Brier scores ranging from 0.0422 to -0.043. The absolute values of calibration-in-the-large ranged from 0.117 to 0.394.

In the testing phase, all methods had reductions in their AUC (range: 0.764 to -0.827) and Brier scores (range: 0.0307 to -0.0336). In the training and validation phases, MLP had the best performance with the highest AUC and lowest Brier score, but in the testing phase, LR was found to have the highest AUC (0.827) and lowest Brier score (0.0307). The absolute values of the external validation calibration-in-the-large ranged from 0.149 to 0.5017, with LR having the lowest value (0.149) and MLP having the highest value (0.5017), showing considerable overfitting. Therefore, the LR model performed reasonably well in the external testing phase with AUC (0.827), Brier score (0.00307), and calibration-in-the-large (0.149). MLP followed, showing better performance than the DT model, with an AUC of 0.816 and a Brier score of 0.003258.

## 4. Discussion

The chi-square test on the drug intoxication mortality predictors showed that mortality statistically significantly differed in most of these variables. Particularly, it significantly differed for the intent predictor in all three phases of the models. This result reflects previous studies from several countries. In the United Kingdom, more than 80,000 people are admitted to an emergency department due to drug intoxication every year, and more than 1,000 die [35]. In the United States, more than 7.3% of the entire 12 year or older population either abuse, or are dependent on alcohol or illegal drugs [36]. According to data of an analysis of the National Emergency Department Information System (NEDIS), the number of suicide attempts admitted to hospitals’ emergency departments registered with the NEDIS data has been consistently on the rise from 2007 to 2010, and drug or pesticide intoxication was the most frequent method [37]. As shown here, intentional drug intoxication accounts for most of the suicide attempt cases in adults, highlighting the need for management.

The results also showed a statistically significant difference in the risk of mortality from drug intoxication among the elderly population (p < 0.01). Ageing diminishes individuals’ physiological abilities and the incidence of trauma among the elderly is on the rise every year, due to medical advances, prolonged life expectancy, and increased participation in social activities. For these reasons, an injury in this population requires more health care services compared to other age groups. Particularly, prognoses are even poorer for patients with chronic diseases with high disease burdens, such as diabetes, coronary artery disease, kidney disease, and lung disease [38]. According to the national-wide survey on the Living Conditions and Welfare Needs of Older Persons, 88.5% of the elderly, aged 65 years or older, have at least one chronic disease [39] and frequently take prescribed and over-the-counter medications. Moreover, various problems, including social, psychological, and health, serve as the cause of drug intoxication among the elderly. In our study, the percentage of drug intoxication, although not fatal, was also high. Hence, as drug intoxication incidences can increase even more with the growing elderly population, it is important to raise awareness on the issue and develop preventive measures for drug management.

Differences in mental disorders were significant in the training phase (p < 0.05), while risk factors were significant in all three phases (p < 0.05). Particularly, psychiatric history, past suicide attempts, family history of suicide, and living alone were identified as risk factors of suicide attempts among intentionally intoxicated patients [40,41]. Therefore, a national-level implementation is needed for evaluating and treating intentional drug intoxicated patients and systematic post-treatment management, such as assessment of psychiatric risk factors, for intentional drug intoxicated patients and their caregivers. In this study, we developed and validated the performance of prediction models for drug intoxication-induced mortality using LR, DT, and MLP. To compare the three models, we processed the samples through training and validation, and tested their performance using the testing dataset. This approach to model testing is infrequent, but methodologically more rigorous than simply considering internal validation.

In the developed prediction models’ validation, the LR model had superior overall performance, discrimination, and external validity in the testing phase compared to the other two models. In addition, all three models had satisfactory overall performance and discrimination, but the DT and MLP had a miscalibration problem in the external validity assessment. Due to this miscalibration, it is possible that applying the DT and MLP models could lead to a systematically incorrect decision when they are applied to new patients. Thus, the models should be updated appropriately to suit the environment in the application stage.

Despite the better data exhibited by MLP, the overall winner between the conventional statistical model and the modern machine learning model is not quite clear for two reasons. First, the MLP and DT demonstrated underwhelming performance in the external validation phase. Second, the complexity of the tuning parameters involved in modern machine learning methods allows for LR, which is less complex, to be perceived as better prediction model. The key finding is that the non-linear and non-additive signals are too weak to make the modern methods beneficial. We suggest that the validation of sophisticated models, such as modern machine learning methods, should involve a comparison with the simple logistic model as a benchmark. Also, using more than one performance measure is essential when comparing the performance of prediction models. For this purpose, the AUC and the Brier score were used in this study. Even though the AUC is a general performance measure, which is limited a discriminator between models, the Brier score, and the calibration in the large are also useful performance indexes when comparing models.

## 5. Conclusions

In this study, a model was derived to estimate the death effects of drug-addicted patients. In particular, we have been predicted using traditional analysis methods (LR) and new analysis methods (DT, MLP). In addition, we compared the usefulness and validity of the model among the techniques to identify a technique suitable for the medical field where accuracy is emphasized. First, we identified the factors affecting the mortality of drug patients through the estimation model and found that similar items were selected as influencing factors in the three models.

Next, the assessment using LR, DT, and MLP to determine the validity of the estimation model resulted in satisfactory results in the model performance and discriminant power. The MLP was found to be superior to other methods in the training and validation phases, but the LT method was found to be more useful in the testing phase. However, in the external validity evaluation, a new technique (DT, MLP) caused an overfitting problem, but LT was relatively superior to the other two techniques in external validity.

That is, although there is a possibility that the new technique can be used to improve the model performance and discriminant, it cannot be said that the new techniques are superior to the traditional ones in the medical field where the overfitting problem is important. This is because the medical field, which is directly related to the patient’s life, has a high standard of accuracy, and, if the estimation model is applied to a new patient, the model is continuously updated to reduce the possibility of making wrong decisions due to overfitting.

In particular, in the medical field, which is directly related to the patient’s life, accuracy is important. So, when applying the estimation model with the problem of overfitting to a new group, it is necessary to modify the model to reduce wrong decision making due to overfitting.

This study has limitations due to the limited clinical indicators and the small sample size of the sample from the panel data, and the use of MLP instead of Random Forests (RF) and Support Vector Machines (SVM), which are advanced models. In addition, there is a limit in reflecting the characteristics of different age groups by dividing the age into two groups based on the age of 65. However, this study has the meaning of predicting the death factor of drug addiction by using data mining techniques and the advantages and disadvantages between the techniques, indicating that consideration should be given in future field applications.

## Figures and Tables

**Figure 1 ijerph-17-00897-f001:**
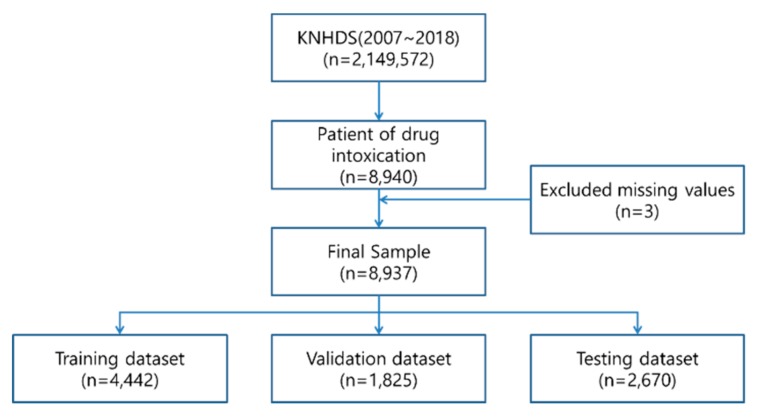
Sample structure.

**Figure 2 ijerph-17-00897-f002:**
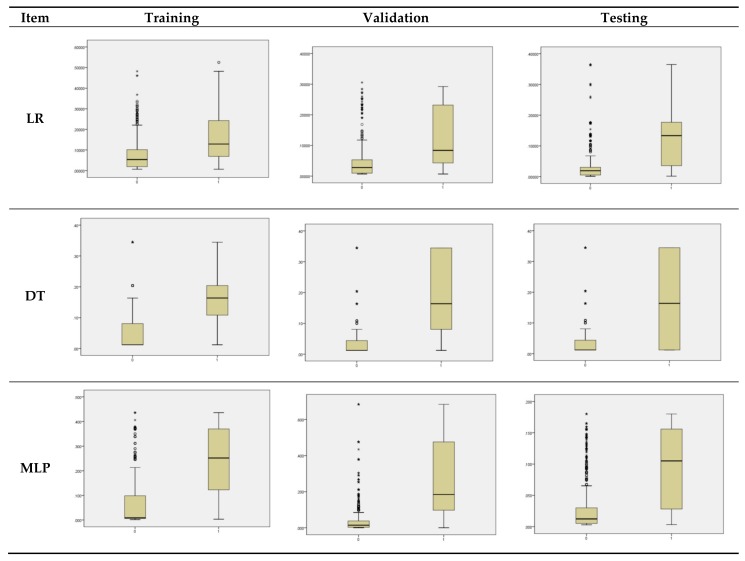
Box Graph.

**Figure 3 ijerph-17-00897-f003:**
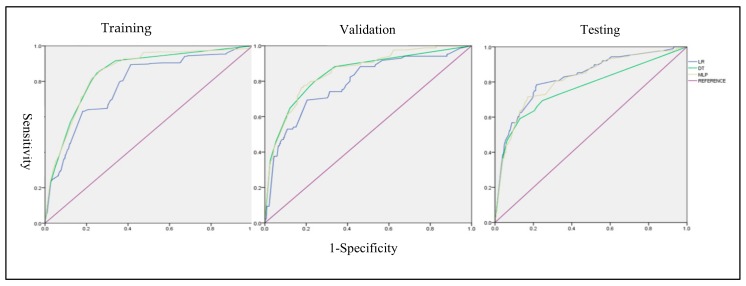
ROC Graph.

**Table 1 ijerph-17-00897-t001:** Results of chi-square.

Items		Training		Validation		Testing	
**Age**	Under 65	3.333(75.0)	**62.1 *****	1.360(74.5)	68.5 ***	1.878(70.3)	67.1 ***
	Over 65	1.109(25.0)		465(25.5)		792(29.7)	
**Toxic Substance**	Toxic Drug	1.768(39.8)	440.1 ***	739(40.5)	104.2 ***	1.176(44.0)	64.4 ***
	Alcohol	28(0.6)		17(0.9)		34(1.3)	
	Hazardous Substance	1.590(35.8)		614(33.6)		809(30.3)	
	Other	1.056(23.8)		455(24.9)		651(24.4)	
**Severity** **(CCI)**	0	3.844(86.5)	17.5 ***	1.577(86.4)	14.5 ***	2.295(86.0)	17.9 ***
	1	403(9.1)		181(9.9)		249(9.3)	
	2	114(2.6)		44(2.4)		73(2.7)	
	3	81(1.8)		23(1.3)		53(2.0)	
**Cause of Risk**	Conflict with Relatives	666(15.0)	11.3 **	192(10.5)	13.3 **	317(11.9)	13.3 **
	Physical Illness	116(2.6)		69(3.8)		94(3.5)	
	Mental Problem	678(15.3)		285(15.6)		327(12.2)	
	Financial Problem	106(2.4)		67(3.7)		114(4.3)	
	Other	2.876(64.7)		1.212(66.4)		1.818(68.1)	
**Intent**	Unintentional	1.657(37.3)	123.8 ***	699(38.3)	25.6 ***	1.021(38.2)	37.3 ***
	Intentional	2.562(57.7)		1.033(56.6)		1.500(56.2)	
	Missing	223(5.)		93(5.1)		149(5.6)	
**Total**		4.442		1.825		2.670	

*** *p* < 0.01, ** *p* < 0.05, * *p* < 0.1.

**Table 2 ijerph-17-00897-t002:** Model performance test.

Items		Brier Score	AUC	Calibration
Logistic Regression	Training	0.06032	0.779	−0.00342
	Validation	0.04266	0.788	0.207416
	Testing	0.030796	0.827	0.149374
Decision Tree	Training	0.060441	0.845	0.244034
	Validation	0.042295	0.845	−0.11715
	Testing	0.033615	0.764	−0.49888
Multilayer Perceptron	Training	0.059971	0.848	−0.31857
	Validation	0.043033	0.853	−0.3938
	Testing	0.032589	0.816	−0.50177

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
