# Peer review of "Comparing Logistic Regression Models with Alternative Machine Learning Methods to Predict the Risk of Drug Intoxication Mortality"

_ijerph, 2020, doi:10.3390/ijerph17030897_

Round 1
Reviewer 1 Report
See attached file

Author Response
Thank you for your kind and detailed review.
Our authors agree with the comments and revise the paper as shown in the attached file.
Notes have been added to the revised section of the article, and you can check the attached revision paper.
Once again thank you for your professional opinion.
I wrote a paper about the English sentence and grammar you pointed out, and I was helped by a professional company. I will attach the relevant proof.
Response to Reviewer 1 Comments
Point 1: Based on the data on Table 1, sex is a feature that is significant at 0.05 in all three phases. Can the authors explain why sex was not included as a feature in the models?
Response 1: In the statistical results created earlier in the study, the gender column did not include gender in the model because it was not statistically significant. As a result, it is not shown in Table 1.
Point 2: The authors adjusted the age variable by dividing the dataset into two groups, those above 65 years of age and those below. Can the authors explain why this choice was made instead to using several different ranges of ages? This choice biases the data by considering all age groups under 65 to be similar.
Response 2: Reviewing the definition of “elderly” by February 2006, Geriatrics and Gerontology International 6(3).
Conventionally, “elderly” has been defined as a chronological age of 65 years old or older, while those from 65 through 74 years old are referred to as “early elderly” and those over 75 years old as “late elderly.”
In the statistics provided by the Korea Centers for Disease Control and Prevention, which were provided with the data set, all the elderly were defined as 65 years of age or older, and all relevant statistical tables were prepared.
Reflecting the current situation, I used the definition of the elderly population as 65 or older.

Reviewer 2 Report
In this paper, the authors developed a drug intoxication mortality prediction model. They compared 3 alternatives for that model, namely Logistic Regression (LR), Decision Tree (DT), and Multi-layer Perceptron (MLP).
The problem is very important, and the authors presented that very thoroughly.
My overall remark is that the current paper is interesting but there is clearly a weakness in the machine learning part.
First of all, neither a description of the employed machine learning models (DT, MLP) is provided nor relevant references.
Moreover, the most crucial issue is the lack of information and transparency regarding the training of the machine learning models. The authors split the dataset into 3 sets (training, validation, testing). However, they do not disclose how they trained the employed models. Usually, machine learning models are validated in a k-fold CV setup and their parameters get tuned in a nested k-fold CV setup, often called internal/inner tuning. Nevertheless, here the datasets are split in training, validation, testing based on chronological order of the samples, which is acceptable. However, it should be clear how the models are trained and how their parameters are tuned. The latter is crucial as even the best machine learning model could lead to terrible results when wrongly tuned or not tuned at all.
Apart from that main concern, below I present my remaining remarks.
The dataset is well presented and described. However, this does not hold when it comes to the evaluation metrics. It is confusing to refer to metrics that are not employed in the study. This should be altered to facilitate the readability and comprehension of the manuscript.
Machine learning approaches such as the DT are not modern. On the contrary, they were proposed many decades ago. Therefore, the authors should alter the corresponding sentences in their manuscript. For example, instead of “modern approaches” they should refer to the applied machine learning methods as “machine learning models recently used in the field” or something like that.
The used MLP model should be described. Is it built in a deep learning architecture? How many layers were used?
DT models are unstable and are considered as relatively inaccurate. The same holds for MLP when trained in a no-deep learning strategy. Thus, it makes sense that LR outperforms them. Comparisons against other models such as Random Forests (RF), or SVM should be included.
Finally, when it comes to machine learning and especially Deep Neural Networks, the existence of many features is often important in order to get an accurate prediction model. This can be a reason behind the low performance of those models.
The conclusion that generally LR should be included as benchmark in regression problems is not new. This is already known in the ML community and often appears in machine learning related studies. The corresponding sentence should be rephrased.
Author Response
Thank you for your kind and detailed review.Our authors agree with the comments and revise the paper as shown in the attached file.
Notes have been added to the revised section of the article, and you can check the attached revision paper.
Once again thank you for your professional opinion.
I wrote a paper about the English sentence and grammar you pointed out, and I was helped by a professional company. I will attach the relevant proof.
I would be grateful if you would give me the opportunity to publish my manuscript in a prominent journal Sincerely yours Yookyung Boo

Round 2
Reviewer 1 Report
The authors have responded to some of my concerns from my previous review while they have not addressed several others. Without their point by point responses to all my concerns, I cannot critically evaluate the paper or recommend the manuscript for publication.
Author Response
Point 1: Based on the data on Table 1, sex is a feature that is significant at 0.05 in all three phases. Can the authors explain why sex was not included as a feature in the models?
Response 1: We extract candidate variables includes sex by previous studies. After, select five variables with an importance index of 10% or higher through importance analysis.
And, Reviewer 2 recommended to reorganize the table 1 after delete unused variables for readability. As a result, it is not shown in Table 1. (see line 82-88)
Point 2: The authors adjusted the age variable by dividing the dataset into two groups, those above 65 years of age and those below. Can the authors explain why this choice was made instead to using several different ranges of ages? This choice biases the data by considering all age groups under 65 to be similar.
Response 2: It is stem from Korea unique condition. South Korea's suicide rate for people aged 65 and over has ranked first among OECD countries since 2009. The number of old people who killed themselves was 54.8 per 100,000, 3.2 times the OECD average.
In particular, suiciding using pesticides of rural residents was a serious social problem. And our research interest to elderly deference, so describe it. (see 203-211)
Even though, there are some weakness as your advice, so describes the research limitation (see line 271).
Reviewer 2 Report
The authors state that: "The best number of hidden units from the range was determined using the testing data criterion." This is still not clear. The authors should elaborate on this testing data criterion or provide a relevant reference.
The authors should also mention the software that was used. Was it python? R? Which package?
My previous comment, namely "Point 3" was not tackled.
My comment was:
The dataset is well presented and described. However, this does not hold when it comes to the evaluation metrics. It is confusing to refer to metrics that are not employed in the study. This should be altered to facilitate the readability and comprehension of the manuscript.
In other words, I think equations referring to measures that are not used in the study (e.g., Concordance Index) should be erased.
line 114: "Discrimination slope, Concordance index (C index), and AUC are used." -> "Discrimination slope, Concordance index (C index), and AUC are often used."
Proofreading of the manuscript is needed as there are still typos. For example:
line 107 the parenthesis "(" does not close.
line 110 model performance of model.. -> model performance.
line 115 two dots were used
MLP stands for Multilayer perceptron not Multilayer
perception.
Author Response
Point 1: The authors state that: "The best number of hidden units from the range was determined using the testing data criterion." This is still not clear. The authors should elaborate on this testing data criterion or provide a relevant reference.
The authors should also mention the software that was used. Was it python? R? Which package?
Response 1: Thank you for your comments. We searched relevant references and include in the article. One is splitting dataset into training, validation, and separated test set. To avoid overfitting and evaluate exact prediction performance (see line 131-133). And, regarding the training of the machine learning models. We searched and added the reference from 37 to 40. And the training and parameters tuning method like as: The MLP trains the network by error back-propagation, and with 1 hidden layer with 9 neurons and hyperbolic tangent activation function was used, and softmax is used as the output layer function for the prediction of diseases [37].(see line 152-162). And we used SPSS 25.0, which is described line 123.
Point 2: My previous comment, namely "Point 3" was not tackled.
My comment was:
The dataset is well presented and described. However, this does not hold when it comes to the evaluation metrics. It is confusing to refer to metrics that are not employed in the study. This should be altered to facilitate the readability and comprehension of the manuscript.
In other words, I think equations referring to measures that are not used in the study (e.g., Concordance Index) should be erased.
line 114: "Discrimination slope, Concordance index (C index), and AUC are used." -> "Discrimination slope, Concordance index (C index), and AUC are often used."
Response 2: Check the sentence and alerted sentence like as : An importance analysis was conducted on these independent variables to identify which should be used in the model comparisons. Based on the results, the following five variables with an importance of 10% or higher, were selected: age, toxic substance, intent of intoxication, severity, and risk factor. (see line 85-95) And deleted equation of Concordance Index. Also changed to "Discrimination slope, Concordance index (C index), and AUC are often used."
Point 3: Proofreading of the manuscript is needed as there are still typos. For example:
line 107 the parenthesis "(" does not close.
line 110 model performance of model.. -> model performance.
line 115 two dots were used
MLP stands for Multilayer perceptron not Multilayer perception.
Response 3: Thank your checking. We checked and revised and the others incorrect words and sentences like as : ~mis~( line 21).
Check the sentence and alerted sentence like as : An importance analysis was conducted
